# Frontal EEG Changes with the Recovery of Carotid Blood Flow in a Cardiac Arrest Swine Model

**DOI:** 10.3390/s20113052

**Published:** 2020-05-28

**Authors:** Heejin Kim, Ki Hong Kim, Ki Jeong Hong, Yunseo Ku, Sang Do Shin, Hee Chan Kim

**Affiliations:** 1Interdisciplinary Program in Bioengineering, Graduate School, Seoul National University, Seoul 03080, Korea; hjkim83@snu.ac.kr; 2Department of Emergency Medicine, Seoul National University Hospital, Seoul 03080, Korea; emphysiciankkh@gmail.com (K.H.K.); emkjhong@gmail.com (K.J.H.); sdshin@snu.ac.kr (S.D.S.); 3Department of Biomedical Engineering, Chungnam National University College of Medicine, 266, Munwha-ro, Jung-gu, Deajeon 35015, Korea; 4Department of Biomedical Engineering, Seoul National University College of Medicine, Seoul 03080, Korea; hckim@snu.ac.kr

**Keywords:** cardiopulmonary resuscitation (CPR), electroencephalogram (EEG), hemodynamic data, carotid blood flow (CBF), cerebral circulation

## Abstract

Monitoring cerebral circulation during cardiopulmonary resuscitation (CPR) is essential to improve patients’ prognosis and quality of life. We assessed the feasibility of non-invasive electroencephalography (EEG) parameters as predictive factors of cerebral resuscitation in a ventricular fibrillation (VF) swine model. After 1 min untreated VF, four cycles of basic life support were performed and the first defibrillation was administered. Sustained return of spontaneous circulation (ROSC) was confirmed if a palpable pulse persisted for 20 min. Otherwise, one cycle of advanced cardiovascular life support (ACLS) and defibrillation were administered immediately. Successfully defibrillated animals were continuously monitored. If sustained ROSC was not achieved, another cycle of ACLS was administered. Non-ROSC was confirmed when sustained ROSC did not occur after 10 ACLS cycles. EEG and hemodynamic parameters were measured during experiments. Data measured for approximately 3 s right before the defibrillation attempts were analyzed to investigate the relationship between the recovery of carotid blood flow (CBF) and non-invasive EEG parameters, including time- and frequency-domain parameters and entropy indices. We found that time-domain magnitude and entropy measures of EEG correlated with the change of CBF. Further studies are warranted to evaluate these EEG parameters as potential markers of cerebral circulation during CPR.

## 1. Introduction 

Approximately 395,000 adults experience an out-of-hospital cardiac arrest (OHCA) annually in the US, and their overall survival rate is only 6–11% [1,2,3]. To prevent death or irreversible damage to vital organs, such as the brain, high-quality cardiopulmonary resuscitation (CPR) is necessary [4,5]. Multiple physiologic measurements have been suggested as indicators of the effectiveness of CPR. End-tidal carbon dioxide (ETCO2) is a widely-used indicator for the pulmonary circulation, and the ETCO2-directed feedback methods are reported to improve the likelihood of return of spontaneous circulation (ROSC) [6].

Recently, achieving good neurological recovery has been regarded as one of the major goals of CPR, because it can influence survivors’ quality of life and their socioeconomic burden [7,8]. However, ETCO2 mainly reflects the systemic circulation, and thus is not adequate to monitor the cerebral circulation or physiological responses of the brain during CPR. Carotid blood flow, the blood supply to the brain, can reflect the cerebral circulation directly. However, its measurement requires an ultrasonic volume flow meter, as well as a skilled operator.

Non-invasive electroencephalography (EEG) can be an alternative to overcome these drawbacks. Portable and low-cost EEG headsets and sensors are currently available out-of-hospital [9]. EEG activity during CPR is reported to be sensitive to cerebral circulation [10,11]. Once cerebral oxygenation decreases due to cardiac arrest (CA), the EEG activity gradually enters the isoelectric state [12,13,14]. However, the EEG activity can return to the pre-arrest state when ROSC was achieved [15,16]. Effective CPR maintains a certain degree of cerebral electrical activity, changing the EEG activity from isoelectric status to large-amplitude and low-frequency status with bispectral index score (BIS) above 40 [17]. As an important tool for determining the prognosis of ischemic episodes of CA patients, EEG signal is routinely monitored for post-resuscitation treatment [18]. The application of EEG monitoring has expanded to the CPR situation, and distinctive EEG patterns are suggested as possible markers for the quality of cerebral resuscitation and oxygen delivery [19]. To date, however, the direct relationship between the carotid blood flow (CBF) recovery and the EEG during CPR has been rarely discussed.

In this study, we focused on the investigation of the relationship between the recovery of carotid blood flow and non-invasive EEG parameters, including time- and frequency-domain parameters, and entropy indices between defibrillation attempts. We applied a single-channel EEG measurement device that was developed in our laboratory and designed a ventricular fibrillation (VF) swine model with simultaneous measurements of EEG and hemodynamic data, including CBF. We hypothesized that CBF recovery may improve cerebral electrical activity, which can result in EEG changes, even during short intervals between defibrillation attempts.

## 2. Materials and Methods

### 2.1. Ethical Statement

The animal test protocol was approved by the Institutional Animal Care and Use Committee of Seoul National University Hospital (IACUC Number: 17-0106). All animal care abided by the Laboratory Animal Act of the Korean Ministry of Food and Drug Safety.

### 2.2. Study Design and Setting

An animal experiment was designed based on a VF swine model. The LUCAS machine (LUCAS2 Chest Compression System, Jolife AB, Sweden) was exploited for mechanical chest compressions (Figure 1a). The machine compressed the chest at a rate of 100 compressions/min with a depth of 5 cm. To prevent displacement of the piston, animals were fixed on the table and the back plate was positioned underneath the animal as a support for the machine. The exact location of the heart was identified by ultrasonic imaging, and then the piston was placed on the site. One emergency medical technician held the machine to prevent the displacement of the piston during CPR.

The assumed scenario of this study was a witnessed OHCA. The duration of untreated VF was determined by considering CA-CALL time (the time of cardiac arrest to call) for bystanders to recognize CA and call emergency services [20]. In addition, it was estimated that four consecutive basic life support (BLS) cycles were performed by the bystanders, with the help of an emergency center, before the dispatched emergency medical team (EMT) arrived at the site. The EMT performed the first defibrillation shock and checked the electrocardiogram (ECG). If the ECG rhythm was shockable, a biphasic defibrillation shock of 200 J was applied by the EMT to restart the heart. Monitoring was initiated when a palpable pulse with organized QRS complexes and systolic blood pressure over 60 mmHg appeared [21]. Sustained ROSC was confirmed if spontaneous circulation continued for 20 min [19]. Once a palpable pulse did not appear after the defibrillation, or VF occurred again during the monitoring session, one cycle of advanced cardiovascular life support (ACLS) was performed by EMT immediately. If the ECG rhythm was still shockable, the defibrillation shock was applied. In case of pulseless electrical activity or asystole, however, the defibrillation shock was omitted, and the next cycle of ACLS was initiated immediately. If a palpable pulse appeared after the defibrillation, then monitoring for 20 min was initiated. Non-ROSC was confirmed if sustained ROSC was not achieved after 10 cycles of ACLS. During ACLS sessions, epinephrine of 1 mg was injected once every 3 min [22]. After the monitoring sessions or all 10 ACLS sessions, the animals were administered euthanasia, with an injection of potassium chloride. Simultaneous EEG and hemodynamic data were collected during the experiments. The entire test scenario, with a timeline, is described in Figure 2.

### 2.3. Experimental Animals and Housing

Eight domestic cross-bred pigs, approximately 3 months of age (45.6 ± 2.4 kg), were studied. The animals were maintained in an accredited Association for Assessment and Accreditation of Laboratory Animal Care (AAALAC) International (#001169) facility, in accordance with the Guide for the Care and Use of Laboratory Animals [23]. They were judged healthy and fasted overnight.

### 2.4. Surgical Preparation and Hemodynamic Measurements

The animals were initially sedated with intramuscular injections of 5 mg/kg of tiletamine hydrochloride and zolazepam hydrochloride (Zoletil, Virbac, France) and 2 mg/kg of Xylazine (Rompun, Bayer, Korea), followed by inhaled isoflurane at a dose of 1–1.5%. Endotracheal intubation was performed on the sedated animals, and a capnography (Capstar-100, CWE Inc., Ardmore, PA, USA) was installed. Mechanical ventilation was initiated. To continue the anesthesia, a tidal volume of 12 mL/kg, respiratory rate of 10 breaths/min, partial pressure of arterial carbon dioxide at approximately 40 mmHg, and partial pressure of arterial oxygen over 80 mmHg were maintained. 

An implantable perivascular probe (MA2PSB, Transonic Systems, Ithaca, NY, USA) combined with a perivascular flowmeter (T420, Transonic Systems, Ithaca, NY, USA) was placed on the internal carotid artery to measure the CBF. A pressure catheter (Mikro-tip pressure catheter, Millar, Houston, TX, USA) was inserted into the left femoral artery and placed in the descending thoracic aorta to measure the arterial blood pressure. Another Mikro-tip pressure catheter was inserted into the right atrium to measure the right atrial pressure. The ECG and saturation of percutaneous oxygen were also measured. All signals except EEG were gathered and saved in a data acquisition system (PowerLab 16/35, ADInstruments, Dunedin, New Zealand) simultaneously.

A pace-making wire was inserted into the right ventricle through the central vein catheter. Isoflurane was stopped before inducing VF to recover EEG signal. EEG started to recover, and appeared similar to the recording obtained before the injections. Then, a direct-current shock was applied to induce VF. Mechanical ventilation was halted, and the animals were left without assistance for 1 min. Thereafter, CPR and defibrillation attempts were executed, and manual ventilation using a resuscitator bag (Ambu Resuscitators, Ambu A/S, Ballerup, Denmark) was initiated to provide positive pressure ventilation to the animal at a rate of once per 10 compressions.

### 2.5. EEG Measurement

A portable single-channel digital electroencephalograph and disposable surface electrodes (MT100, Kendall Healthcare, Toronto, Ontario, Canada) were attached to measure the scalp EEG under the referential montage. Reference and ground electrodes were attached on either side of the mastoid. Active electrodes connected to the device were placed on the forehead (Figure 1b). The raw EEG signal was bandpass filtered with a frequency range of 0.5–47 Hz and amplified with a gain of 12,000 *v*/*v*. The amplified signal with a low noise level under ±3 µVp-p was digitized and transmitted to the laptop via Bluetooth communication at a rate of 250 Hz. The data acquisition software in the laptop receives and saves the EEG data.

### 2.6. Data Processing

All data were processed using MATLAB (MATLAB R2017b, Mathworks, Natick, MA, USA). The EEG and hemodynamic data were synchronized. Approximately 3-s-long pauses right before the defibrillation shocks were selected for analysis. The selected EEG was segmented into three 2-s-long sub-epochs with 1.5-s overlaps to reduce variation; 0–2 s, 0.5–2.5 s, and 1–3 s period. The representative EEG parameters were obtained from the average of three sub-epochs. Segmenting the EEG and obtaining parameters is similar to the signal processing technique for the BIS monitor [24]. Time and frequency domain parameters and entropy indices were obtained in this manner. All parameters considered are listed in Table 1.

### 2.7. Data Analysis

First, CBF recovery during CPR were analyzed to investigate their relationship with resuscitation rates. The recovery rate was defined as a relative scale of each hemodynamic parameter with respect to the baseline value in the pre-VF state. Second, the EEG waveforms were scrutinized according to the test scenario. EEG activity was evaluated, along with the recovery of CBF.

Pearson correlation coefficients between each EEG parameter and the recovery rates of CBF for all experiments were obtained to inspect whether EEG parameters show similar changes with the CBF. In addition, the recovery rates of CBF were categorized into four quartile groups: group 1 (<25%); group 2 (25–50%); group 3 (50–75%), and group 4 (>75%). Averages of each EEG parameter among groups were evaluated through one-way analysis of variance (ANOVA). Significance was considered at a level of *p* < 0.05. Receiver operating characteristic (ROC) curve analysis was also performed to measure the optimal cut-off values of EEG parameters, to discriminate between the higher and the lower group of the CBF recovery based on the median value, which was approximately 30%. These tests were performed with SPSS (SPSS Statistics 23, IBM SPSS Statistics, New York, NY, USA).

## 3. Results

### 3.1. Results of CPR Process

All eight experiments were performed successfully. Once VF was triggered, mean arterial pressure (MAP) decreased to almost 0 mmHg, while approximately 20% of baseline MAP remained as residual pressure in the vessel. CBF dropped rapidly to almost 0% of baseline values during untreated VF. When BLS sessions began, hemodynamic parameters started to recover. Recovery rates of hemodynamic parameters over the BLS and ACLS sessions are presented specifically in Appendix A.

Sustained ROSC was achieved in five animals. Among them, one animal was defibrillated after the last BLS session. Another four animals were defibrillated during the course of the ACLS sessions. No animals experienced VF again during the monitoring sessions. Three animals were not resuscitated until the tenth ACLS session was completed. BLS cycles were performed a total of 32 times, and ACLS cycles were performed 48 times, and data after those sessions were included for analysis.

### 3.2. EEG Changes with the Recovery of CBF

The EEG waveforms between an ROSC (Test 6) and a non-ROSC (Test 5) case were compared (Figure 3). Before VF, the amplitude of EEG with irregular morphology exceeding ±20 µV was observed. Since cerebral oxygenation decreased due to VF, the amplitude started to decrease in 10–15 s and almost entered the isoelectric state (±5 µV) at the end of untreated VF.

The recovery of the EEG was different, depending on the recovery of the CBF. In Test 6, which showed a better recovery, the recovery rate reached almost 40% during the last two BLS sessions. Concurrently, an increased background activity with higher amplitude and increased higher frequency components was observed. EEG activity during the monitoring session appeared similar to the baseline values during the pre-VF period. This means that the cerebral circulation was restored successfully, whereas the recovery rates in Test 5 exceeded 30% during the first BLS session but decreased consistently during the rest of the CPR sessions. The EEG decreased in amplitude and entered the suppression status and increased lower frequency components during the second BLS session. The cerebral resuscitation was poor, with the low CBF recovery rates of below 10%. Nearly flat patterns resulting from electrocerebral inactivity appeared, and EEG did not recover until the end of the ACLS sessions.

Table 2 shows the Pearson correlation coefficients between EEG parameters and the recovery rates of CBF. Among them, time-domain magnitude and two entropy indices, log energy entropy [25] and Rényi entropy [26], showed a correlation coefficient of approximately 0.78. Figure 4 demonstrates the scatter plots for these three parameters. 

### 3.3. Changes in EEG Parameters Depending on Four CBF Groups

Figure 5 illustrates the results of one-way ANOVA tests for three parameters. For magnitude, the lowest quartile (group 1) showed significant differences to other groups, with *p* < 0.05. However, significant difference was not confirmed among the other three groups. Similar patterns were observed in following two entropy indices. Table 3 demonstrates the results of the post hoc test based on the Dunnett T3 method.

### 3.4. EEG Parameters Depending on the Different CBF Recovery Groups

Figure 6 illustrates the ROC curves for the three EEG parameters. All possible cut-off values are plotted with a combination of true positive rate (sensitivity) and false positive rate (1 − specificity). The optimal cut-off points are also denoted. Table 4 presents the results of ROC curve analysis including area under the curve (AUC), true positive rate, false positive rate, and cut-off values. The AUC values of all three parameters were over 0.88.

## 4. Discussion

This study investigated the relationship between the EEG and CBF, to evaluate the feasibility of non-invasive EEG parameters as potential predictors of the recovery of CBF in the CA swine model. The current CPR protocol consists of an ECG rhythm check, chest compression (CC), defibrillation, and medication [22], while CBF or EEG measurement and analysis have rarely been performed during CPR. Monitoring cerebral circulation could provide beneficial information to improve patients’ prognosis and quality of life [7,8]. EEG was considered as one of the possible markers because it could reflect the level of cerebral circulation [27]. Post-resuscitation care could be seriously disrupted with a sparsity of EEG activity [28]. If the EEG could reflect the CBF and be measurable in the OHCA setting, CPR with a feedback of non-invasive EEG parameters could guide EMTs to achieve a higher CBF recovery, for example, by guiding leg elevation or the Trendelenburg position [29], which is expected to improve brain perfusion and neurologic outcomes of CA patients after CPR. It is noteworthy that the present study used only single-channel EEG signals from forehead sites where the installation of EEG sensors is convenient.

Several studies have attempted to apply the BIS monitor during CPR. However, unwanted artefacts due to CCs contaminated the original EEG, and generated unreliable outputs [30,31]. The BIS monitor is not adequate to use for the short intervals between CCs, because it is based on the moving-average function over 60 s [24]. Prolonged no- or low-flow periods can deteriorate brain function of CA patients [19,32]. Thus, this study focused on data measured during short pauses between the defibrillation attempts. We observed that the EEG background activity increased and became more irregular with the CBF recovery. The frequency distribution of EEG was also affected. As the CBF recovered, the higher frequency components including alpha (8–13 Hz) and beta (13–30 Hz) increased, whereas the lower frequency components including delta (<4 Hz) and theta (4–8 Hz) decreased. These changes affected the functional dynamics associated with varying amplitudes and multi-frequency responses, including the level of complexity and the amount of energy, diversity, and randomness [33], which could be indicated by the increase of log energy entropy and Rényi entropy, as shown in Figure 4. Entropy parameters have been applied to EEG signals, especially in anesthesia or epileptic seizure studies [34,35,36]. A previous study analyzing epileptic EEG signals reported that log energy entropy of the modulated EEG signals obtained from the epileptogenic area had relatively lower values [35]. Consistently, another study showed that the complexity derived by Rényi entropy was also higher in healthy signals [36]. These parameters might also have a potential to identify sufficient cerebral circulation for satisfying the metabolic requirements of brain cells of CA patients [27].

This study has several limitations. First, the experimental model was finalized assuming a witnessed OHCA. EEG parameters, such as log energy entropy and Rényi entropy, might be able to reflect the cerebral resuscitation only with very short no- or low-flow duration (<1 min). The association between the CBF and EEG recovery is probably less pronounced with a longer untreated VF. Further research should be performed to validate this method with a longer VF period for at least 5 min. Second, this study was performed only with the limited sample size of eight animals. Feature analysis with larger datasets should be performed to confirm our findings. To generalize our findings to real OHCA patients, moreover, future clinical studies should be guaranteed with the experimental setup optimized for human anatomy.

## 5. Conclusions

We measured a single-channel EEG non-invasively during CPR and evaluated the relationship between EEG parameters and the CBF recovery. Our findings indicated that time-domain magnitude and entropy indices of EEG, even during the brief pause in CPR, may correlate with the level of cerebral circulation. Further studies are warranted to evaluate these parameters as potential markers of cerebral resuscitation.

## Figures and Tables

**Figure 1 sensors-20-03052-f001:**
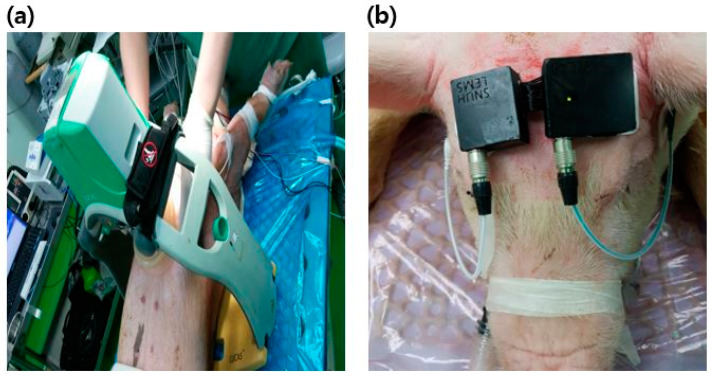
Experimental setup: (**a)** LUCAS2 chest compression system installed on the chest of animals; (**b**) A single-channel electroencephalography (EEG) device mounted on the forehead.

**Figure 2 sensors-20-03052-f002:**
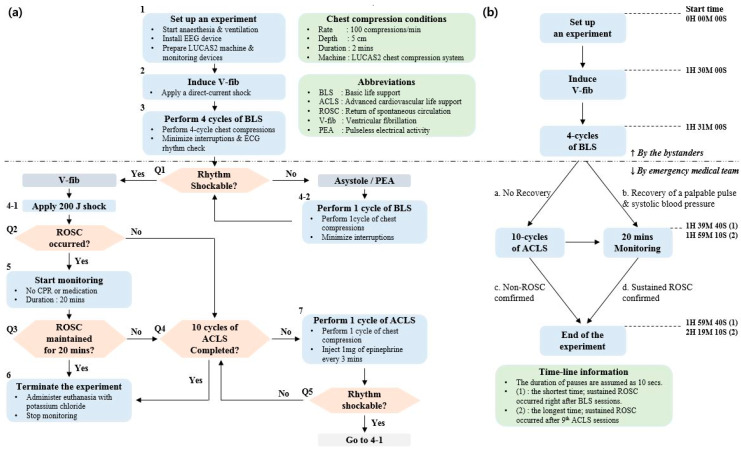
The entire test scenario: (**a**) flow chart from surgical procedure to basic life support (BLS), advanced cardiovascular life support (ACLS), and termination; (**b**) brief timeline of the test protocol.

**Figure 3 sensors-20-03052-f003:**
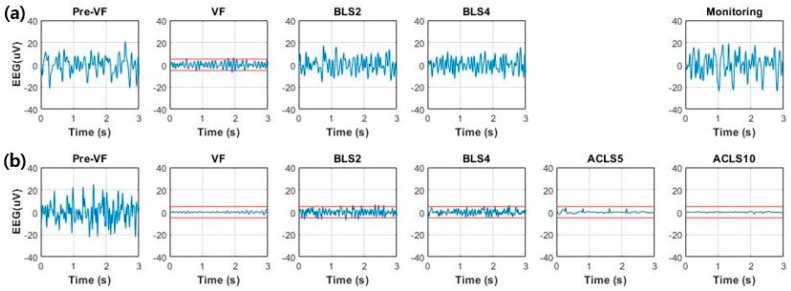
Comparison of EEG over time between return of spontaneous circulation (ROSC) and non-ROSC cases: (**a**) EEG waveforms obtained from animals with successful defibrillation after fourth BLS session and sustained ROSC confirmed after follow-up monitoring for 20 min (Test 6); (**b**) EEG waveforms obtained from animals in which ROSC was not achieved until the end of experiment (Test 5). Dashed lines denote the level of ±5 µV, the limits of the isoelectric state.

**Figure 4 sensors-20-03052-f004:**
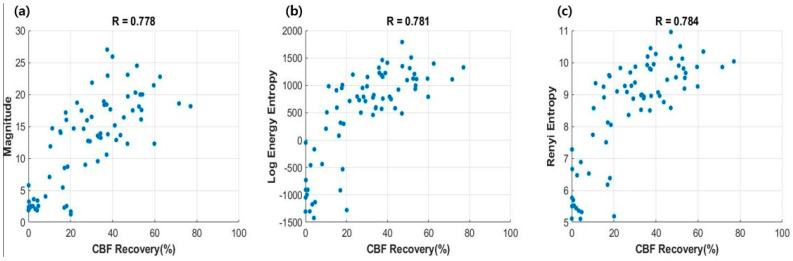
Scatter plots between EEG parameters and the recovery of CBF. Correlation coefficients were denoted above the plots: (**a**) magnitude; (**b**) log energy entropy; (**c**) Rényi entropy.

**Figure 5 sensors-20-03052-f005:**
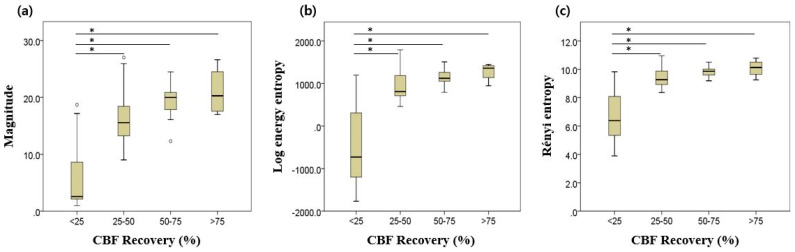
Results of one-way ANOVA: (**a**) magnitude; (**b**) log energy entropy; (**c**) Rényi entropy. Asterisk (*) denotes statistical significance at the *p* < 0.001 level. Error bars indicate the upper and lower extreme values of the data.

**Figure 6 sensors-20-03052-f006:**
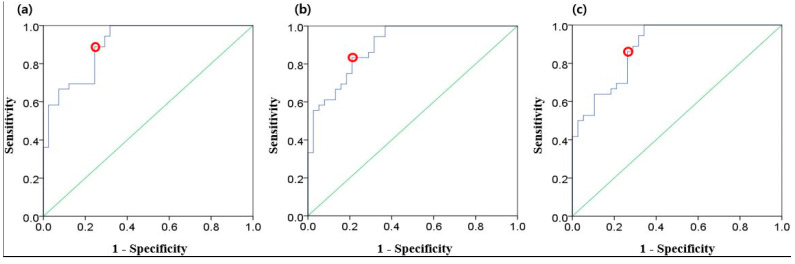
Receiver operating characteristic (ROC) curves (blue) for three EEG parameters: (**a**) magnitude; (**b**) log energy entropy; (**c**) Rényi entropy. Red dots indicate the optimal cut-off points, and the diagonal lines (green) indicate random chance.

**Table 1 sensors-20-03052-t001:** EEG parameters considered in this study.

EEG Parameters	Definition	Domain
Magnitude	Maximal Amplitude during the Epoch (unit: µV)	Time
SynchFastSlow	log(B_0.5–47 Hz_/B_40–47 Hz_)	Frequency
BetaR	log(P_30–47 Hz_/P_11–20 Hz_)	Frequency
DeltaR	log(P_8–20 Hz_/P_1–4 Hz_)	Frequency
AlphaPR	P_8–13 Hz_/P_0.5–47 Hz_	Frequency
BetaPR	P_13–30 Hz_/P_0.5–47 Hz_	Frequency
DeltaPR	P_0.5–4 Hz_/P_0.5–47 Hz_	Frequency
ThetaPR	P_4–8 Hz_/P_0.5–47 Hz_	Frequency
BG_Alpha+	P_8–47 Hz_/P_0.5–47 Hz_	Frequency
Log energy entropy	$\sum_{i=1}^{n}\log{\left(p\left(x_{i}\right)\right)}^{2}$	Entropy
Rényi entropy	$\frac{1}{1-\alpha }\log(\sum_{i=1}^{n}p{\left(x_{i}\right)}^{\alpha })\;,\;\left(\alpha \;\geq 0,\;\neq 1\right)$	Entropy

Abbreviation: P_a__–b Hz_, the sum of spectral power from a–b Hz; B_a__–b Hz_, the sum of bispectral activity from a–b Hz; $p\left(x_{i}\right)$, probability distribution function of signal $x_{i}$; α of Rényi entropy was 0.5.

**Table 2 sensors-20-03052-t002:** Pearson correlation coefficients between EEG parameters and the recovery rates of CBF.

EEG Parameters	Correlation Coefficient	*p*-Value
Magnitude	0.778	<0.001
SynchFastSlow	0.210	0.228
BetaR	−0.329	0.016
DeltaR	0.196	0.032
AlphaPR	0.189	0.048
BetaPR	0.323	0.001
DeltaPR	0.032	0.797
ThetaPR	−0.354	0.004
BG_Alpha+	0.262	0.006
Log energy entropy	0.781	<0.001
Rényi entropy	0.784	<0.001

**Table 3 sensors-20-03052-t003:** Results of multiple comparisons between groups in three EEG parameters.

		Magnitude	Log Energy Entropy	Rényi Entropy
Group I/ Group II		Mean Difference/ Standard Deviation (*p*-value)	Mean Difference/ Standard Deviation (*p*-value)	Mean Difference/ Standard Deviation (*p*-value)
1	2	−10.39/1.24 (<0.001)	−1375.15/164.65 (<0.001)	−2.69/0.319 (<0.001)
1	3	−13.34/1.37 (<0.001)	−1590.42/164.87 (<0.001)	−3.13/0.321 (<0.001)
1	4	−15.15/2.39 (0.012)	−1720.80/190.02 (<0.001)	−3.38/0.434 (<0.001)
2	3	−2.95/1.30 (0.169)	−215.27/87.24 (0.108)	−0.442/0.171 (0.084)
2	4	−4.75/2.35 (0.395)	−345.65/128.60 (0.180)	−0.695/0.338 (0.384)
3	4	−1.80/2.41 (0.958)	−130.39/128.87 (0.871)	−0.253/0.340 (0.957)

Differences were obtained by Group I minus Group II.

**Table 4 sensors-20-03052-t004:** Results of the ROC curve analysis for EEG parameters.

EEG Parameter	AUC	Standard Error	True Positive Rate (Sensitivity)	False Positive Rate (1-Specificity)	Cut-off Value
Magnitude	0.904	0.033	0.889	0.244	12.802
Log energy entropy	0.896	0.035	0.833	0.211	739.543
Rényi entropy	0.885	0.037	0.861	0.263	8.919

Abbreviation AUC: Area under the curve.

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
