# Peer review of "Frontal EEG Changes with the Recovery of Carotid Blood Flow in a Cardiac Arrest Swine Model"

_sensors, 2020, doi:10.3390/s20113052_

Round 1

Reviewer 1 Report

Dear Editor,

Thank you very much for the opportunity to review this paper.

This paper has five major limitations:

  • The duration of untreated VF of 1 minute is very short, too short. Most study use longer duration of VF 5 minutes up to 15 minutes.
  • The EEG part of the study has a major bias. The increases of EEG parameters are probably due to a better hemodynamic. It means that ROSC is due to the CPR component of resuscitation and not the EEG itself. The CPR was more efficient (increase CoPP and carotid blood flow) this result increases the probability of ROSC and in the same time, it improves the EEG. They did not prove that the EEG itself, that can be assimilated as an increase brain function, has a direct effect on the heart aka ROSC. The way some part of paper is written gave the impression that the EEG has an effect of ROSC while in other parts, it is clearer that is due to hemodynamic.
  • Two CPR device are used for this study, which has a very limited number of swine. Even if the hemodynamic seem to be the same, this also creates a bias. The same device should have been used for every swine.
  • Isoflurane was not stopped for 1-2 minutes before VF was induced to wash out the isoflurane. This is performed in a lot of study to remove an anesthetic bias. In this study with a very short VF that uses EEG as an endpoint, this introduces another major limitation.
  • The literature search about EEG/BIS and CPR is basically absent from the discussion (see bellow).

“Our study showed that non-invasively measured EEG parameters may have a potential to predict defibrillation success.” This is an extrapolation. EEG shows a better brain perfusion due to a better CePP due to better hemodynamic in CPR, which at the same time will increase chance of ROSC in a swine that had only an untreated VF of 1 minute. If VF was longer, they would be more brain injuries and a worse EEG. We are able to bring back patient who are brain dead. In those patients, it is unlikely that EEG would have predicted ROSC. This study tries to correlated ROSC and EEG. In other terms CoPP and CePP. They are correlated because if we oversimplify, they are both correlated to MAP. However, if we tied  the carotid CePP would be nulle and CoPP would be normal; EEG would be bad but we could still have ROSC. This study does a syllogism: All men are mortal. Socrates is a man. Therefore, Socrates is mortal. In this case Hemodynamic increase ROSC, EEG increase with hemodynamic, ROSC increase with hemodynamic.

The paper could be changed to EEG parameters as potential predictors of brain perfusion/ CPR quality and remove the ROSC part.

Best regards,

INTRODUCTION

“EEG activity during CPR were reported to be highly sensitive to spontaneous cerebral circulation” and “pre-arrest state when spontaneous circulation ” the word spontaneous give the impression that flow happened by itself while it is due to CPR, and it should be removed.

“Monitoring the EEG recovery during CPR has potential predictive value” The reference 18 is a case report. It does not state that EEG (in the case of the case report BIS monitoring) has a predicted value. The authors are extrapolating.

MATERIAL

“CA-CALL” the authors should define this acronym.

The authors should add a time line figure in the figure 2 of the study.

“Unless sustained ROSC was achieved, one cycle of advanced cardiovascular life support (ACLS) was performed followed by defibrillation”. The authors define sustained ROSC as ROSC for 20 minutes. This phrase should be re-written as I doubt that they wait 20 minutes to perform the next round of CPR.

“Non-ROSC was confirmed if sustained ROSC was not achieved after 10 cycles of ACLS.” Are the rounds of CPR performed by the bystanders in those 10 rounds of ACLS, by definition bystander performed BLS? Again, as the authors used bystanders then EMT then ACLS, a time line of the study is necessary.

According to the figure 2; defibrillation was done at 360j. Even if it is possible to do defibrillation with biphasic defibrillation at that level, the guideline and most swine studies use 120 to 200j. This is also due to the weight of the swine. If they used a monophasic defibrillator where 360j makes more sense, this added another problem to the study as monophasic defibrillators are not recommended any more.

“Isoflurane was injected to anesthetise the animals through the central vein catheter inserted to the right internal jugular vein.” I did not know that Isoflurane could be injected IV. Could the authors give a reference about this?

“A pace-making wire was inserted into the right ventricle through the central vein catheter, and a direct-current shock was applied to induce VF. Mechanical ventilation was halted, and the animals were left without assistance for 1 minute.” As explain above; Had isoflurane been stopped for 1-2 minutes before VF was induced to wash out the isoflurane. This is performed in a lot of study to remove an anesthetic bias. In this study with a very short VF that uses EEG as an endpoint, this introduces another limitation.

RESULTS

“The MAP decreased to 22.1% of the baseline on average during VF” I have performed several hundreds of cardiac arrest study in swine. MAP (Mean Arterial Pressure = 1/3(SBP) + 2/3(DBP)) should decrease to basically 0 during untreated VF as there is no systole and diastole. There is only residual pressure in the vessel due to the pressure of the system.

Figure 3. The ECGs of those two animals are also clearly different. An AMSA study would show a difference. This is also an element in favor that ROSC was obtained due to the hemodynamic parameter been different between both groups.

“Table 1 also demonstrates the results of GEE”. There is a typo at GEE/EEG.

DISCUSSION

There are multiple studies with BIS during CPR. The authors describe their EEG analysis as a “process is similar to the signal processing technique for the BIS monitor “. It is surprising that the discussion does not discuss those papers. A 60 seconds google search of BIS and CPR found:

  • doi: 10.1007/s12245-008-0037-z
  • doi: 10.4097/kjae.2013.64.1.69

The authors need to add this section about EEG/BIS during CPR and update the literature search.

Reviewer 2 Report

Thank you for giving me a chance to review your great work. You researched the relation between haemodynamics and EEG patterns in swine cardiac arrest model. I'd like to give you some comments on your paper.

  1. As you know, EEG can't be applicable during BLS in OHCA. Therefore data analysis for verifying the relation between hemodynamics and EEG must include ACLS data only.
  2. I can't find the reason or reference why you defined the ROSC group as CBF≥30%. You'd better to describe why you categorize the groups by 30% of CBF. Cerebral resuscitation can not be regarded as ROSC.
  3. Different mechanical CPR device can make different hemodynamics and it can be bias for analyzing resuscitation parameters, even if the parameters for CPR quality, e.g. chest compression depth or rate, would be similar. Please describe why you use two types of mechanical CPR device for obtaining same parameters.
  4. It would be better performing receiver operating curve analysis and determining cut-off values for hemodynamic (e.g. MAP, COPP, CBF etc.) and EEG parameters (BSR, Ratio>10, etc.). In general, recovery rate and CBF are not presented as categorical variables.
  5. (Line 206) EEG and ECG patterns in selected cases have little clinical information

  1. (Line 98) In general, defibrillation is performed with biphasic 200 J in clinical and experimental environment. Please show why you choose a biphasic 360 J for defibrillation.
  2. (Fig 2) In recent CPR guidelines, administration of epinephrine is recommended  every 3-5 minutes. Please describe why you decide to select above protocol. And please present absolute amount (mg) of epinephrine rather than solution (cc).
  3. (Line 119) Isoflurane is a inhalation anesthetics so that it can not be administered via central vein. Please revised the anaethetic methods.
  4. (Line 134) Please describe CPR progress more in detail and change the market product name, ambu, for general term.
  5. (Line 154) Please add a brief description for clinical meaning of each EEG parameters.

Round 2

Reviewer 1 Report

Dear Editor,

Thank you very much for the opportunity to review the revised version this paper.

As explain in my initial evaluation this paper have several major limitations.

The authors were able to justify/correct several of them:

  • Isoflurane stopped before VF.
  • The literature search about EEG/BIS
  • The endpoint of the study was change and now focus more on the relation between EEG and cerebral perfusion (represented by CBF). The authors did a good job of removing the predicting value of EEG of ROSC.

However, there is still 2 limitations that cannot be correct. However, with the major change done to the document they have a minimal impact on the study.

  • The duration of untreated VF of 1 minute is very short, too short. Most study use longer duration of VF 5 minutes up to 15 minutes. In the actual presnentation of the study which has remove the ROSC element of the study to focus on the hemodynamic this has a low impact.
  • Two CPR device are used for this study, which has a very limited number of swine. Even if the hemodynamic seem to be the same, this also creates a bias. The same device should have been used for every swine.

At this point no more correction are necessary to the document.

Best regards,

Author Response

Please see the attachment. Thank you for your valuable comments.

Reviewer 2 Report

Thank you for your great effort for revising a manuscript following my comments.

I'd like to suggest some concerns your revised manuscript.

  1. You described that one of the reason using different mechanical CPR device was to test feasibility of KNU-CPR actuator. It is an important principle of research that two or more different studies should not be conducted together on one object. You'd better to use data from one of mechanical CPR device to keep this principle.
  2. It is hard to support your major discussion point that CBF and several factors of EEG were positively correlated with low correlation coefficiency even though the categorical CBF group showed the difference statistically. As I recommended in previous review, ROC curve analysis and presenting cut-off value of each EEG parameters would be more informative to readers.

Author Response

(The authors gave the same response as above.)
